# Gait Analysis with an Upper Limb Prosthesis in a Child with Thrombocytopenia–Absent Radius Syndrome

**DOI:** 10.3390/jcm14072245

**Published:** 2025-03-25

**Authors:** Sebastian Glowinski, Sebastian Pecolt, Andrzej Błażejewski, Igor Maciejewski, Tomasz Królikowski

**Affiliations:** 1Institute of Health Sciences, Slupsk Pomeranian Academy, Westerplatte 64, 76200 Slupsk, Poland; 2Institute of Physical Culture and Health, State Higher School of Vocational Education in Koszalin, Lesna 1, 75582 Koszalin, Poland; 3Faculty of Mechanical Engineering and Power Engineering, Koszalin University of Technology, Sniadeckich 2, 75453 Koszalin, Poland; sebastian.pecolt@tu.koszalin.pl (S.P.); andrzej.blazejewski@tu.koszalin.pl (A.B.); igor.maciejewski@tu.koszalin.pl (I.M.); tomasz.krolikowski@tu.koszalin.pl (T.K.)

**Keywords:** absent radius, gait analysis, prothesis, TAR syndrome

## Abstract

**Background/Objectives**: Thrombocytopenia–absent radius (TAR) syndrome is a rare genetic disorder characterized by the bilateral absence of the radius and thrombocytopenia, often leading to functional limitations and gait asymmetries. Prosthetic devices are sometimes employed to improve mobility and posture, but their impact on gait mechanics in pediatric patients remains poorly understood. **Methods**: The methodology used is based on a study that evaluated the gait parameters of a 10-year-old child with TAR syndrome under static and dynamic conditions, both with and without the use of a custom-designed upper limb prosthesis. The analysis focused on assessing the prosthesis’s impact on gait symmetry and biomechanics. A key aspect of the methodology involved studying the distribution of pressure forces on the ground during walking using the FreeMed EXTREME Maxi baropodometric platform. **Results**: Gait analysis demonstrated asymmetries between the left and right feet. In the absence of the prosthesis, the patient exhibited excessive forward loading and uneven pressure distributions. The use of a custom prosthesis, particularly with counterbalancing features, improved gait symmetry but led to increased reliance on the left foot. This foot experienced higher pressures (738–852 g/cm^2^) and longer ground contact times (690–865 ms) compared to the right foot (619–748 g/cm^2^ and 673–771 ms). The left foot displayed elevated forefoot pressures (61–65%), while the right foot bore weight laterally (66–74%). **Conclusions**: The custom prosthesis influenced gait mechanics by redistributing plantar pressures and modifying ground contact times, partially improving gait symmetry. However, compensatory strategies, such as increased loading on the left foot, could contribute to musculoskeletal strain over time. Individualized rehabilitation programs and prosthetic designs are essential for optimizing gait mechanics, improving mobility, and minimizing long-term complications in TAR syndrome patients.

## 1. Introduction

Thrombocytopenia–absent radius (TAR) syndrome is a congenital malformation syndrome characterized by the bilateral absence of the radii and thrombocytopenia [1]. It was first described in 1929, but it was not until 1969 that it was clearly defined as a syndrome. The estimated frequency of this syndrome is 0.42 cases per 100,000 live births [2].

The lower limbs and gastrointestinal, cardiovascular, and other systems may also be associated with TAR [3]. An additional inherent feature of TAR syndrome is thrombocytopenia. This condition prevents normal blood clotting. It leads to easy bruising, frequent nosebleeds, and bleeding from the gums and mucous membranes, as well as bleeding even after the slightest injuries [4]. Research into the causes of the disease has been conducted for many years [5,6]. The causes of thrombocytopenia range from benign conditions, such as drug-induced or infection-related ones, to more severe conditions, including bone marrow disorders and autoimmune diseases [7,8]. Understanding the underlying mechanisms is crucial for appropriate management and treatment strategies [9]. TAR syndrome significantly affects individuals’ upper limb length and structure, influencing daily activities and gait parameters. This rare congenital condition involves the bilateral absence or deformity of the radius bones, often with additional skeletal anomalies such as shortened ulnae or humeri. These musculoskeletal challenges lead to compensatory gait adaptations, including altered stride length, slower walking speeds, and deviations in balance due to compromised arm movement and its stabilizing role.

An independent task, and at the same time a challenge, is the identification of gait pathologies, on the one hand, and the development of methods and devices for the rehabilitation of people affected by such syndromes, on the other. These two tasks are closely related. In this context, the most well-known, widespread, and cost-effective approach is the identification of gait pathologies, which includes the analysis of kinematics and dynamics, i.e., recording and measuring biomechanical parameters of gait. Various methods are used to record these parameters. The registration of kinematic parameters includes video cameras and motion capture systems [10,11]; photogrammetry [12], i.e., the analysis of movement based on serial images; optical systems with or without markers; accelerometers [13,14]; MEMS sensors [15]; and gyroscopes [16,17], which record accelerations and rotations in real time. The registration of dynamic parameters involves technologies such as force plates, which measure ground reaction forces during gait [18]; dynamometers—devices that measure the force generated by muscles [19]—and pressure sensors, such as pressure mats, which record foot pressure distribution [20,21]. Based on the measurement and recorded biomechanical parameters, various scientific methods can be applied to identify deviations from the norm, asymmetry, imbalance, and more. These methods include signal analysis [13,22], image analysis [23], statistical analyses [24], and, finally, artificial intelligence techniques [25,26]. Individuals with TAR syndrome often exhibit asymmetrical gaits or require adaptive devices for mobility. The absence of full upper limb functionality disrupts typical swing phase mechanics, which can influence energy efficiency and postural stability. Studies highlight how these adaptations increase the metabolic cost of walking and the risk of falls, underlining the importance of tailored physiotherapy and assistive technology. Moreover, the altered biomechanics observed in TAR patients contribute to changes in muscle use and joint load distribution in the lower extremities. This can lead to secondary complications such as joint pain or premature wear in the hips and knees. Ongoing research emphasizes the necessity for early intervention and individualized therapy to improve mobility outcomes and reduce long-term orthopedic complications.

Properly diagnosed people require cyclical and sometimes permanent rehabilitation to support their functioning. Here, cheap, safe rehabilitation devices can be used along with an appropriate set of movement exercises, which is the biggest challenge. For example, a system based on elastic cords that are attached to the patient’s body with harnesses or belts and to a special rigid frame [27,28] allows for movements in controlled relief, supports motor response, and improves postural control. This system also allows for the precise adjustment of the level of resistance or relief, which makes the therapy more individual and effective [29,30,31].

The aim of our study was to evaluate the gait parameters of a child with TAR syndrome. Subsequently, the patient was fitted with an active prosthesis specifically designed for them. The type and frequency of deviations in dynamic gait parameters were determined across different configurations.

## 2. Materials and Methods

### 2.1. Participant

A 10-year-old patient with congenital upper limb deformity was evaluated (Figure 1). In the posture analysis, the following observations were made:-Frontal plane: The head was positioned correctly, shoulders were level, waist angles were symmetrical, the line of spinous processes showed no lateral deviations, and the alignment of the knees and feet was within normal limits.-Sagittal plane: The head was positioned correctly, shoulders were in protraction, and there was a slight anterior pelvic tilt.

**Figure 1 jcm-14-02245-f001:**
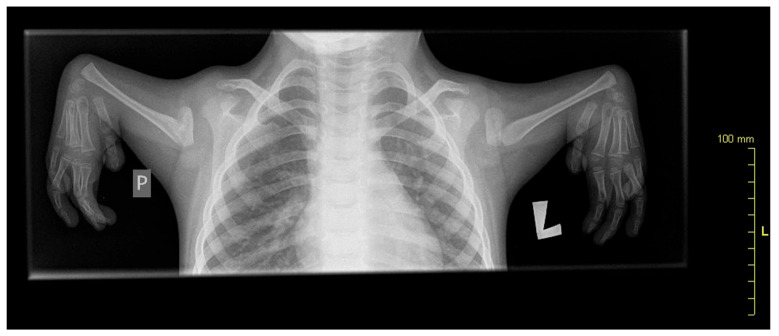
X-ray showing radial aplasia. Note that the bones of the hand appear normal.

In activities involving basic locomotor functions, the patient does not significantly differ from peers in the same age group. The child can assume all age-appropriate positions and shift body weight in all directions. In the quadruped position, the patient uses wrist support with the hand flexed. They perform axial rotation and all phases of transitioning from low to high positions without difficulty.

Pathologies in the malformed upper limbs partially prevent standard measurements of active and passive range of motion. However, observations reveal no significant limitations in passive and active shoulder joint movement, with the exception of an elbow flexion contracture. In the wrist joint, the patient exhibits limited extension but normal flexion. In the metacarpophalangeal and interphalangeal joints, there is full passive extension and flexion, while active flexion is limited. The thumb is in an adducted position, with partial opposition and flexion possible. In the functional “Precision Grip Test”, he can perform the following grips: “hammer”, “hook”, and a grip to turn a handle and open a door. He encounters partial difficulties with precision grips: the “pinch grip” used for turning a key and the “pinch grip” used for picking up coins from a table. In the “Test of Precise Hand Use I”, the child is able to write, and their handwriting is legible and neat. The primary functional issue reported by the individual is difficulty in lifting and holding heavier objects, as well as manipulating them in space.

### 2.2. Upper Limb Prosthesis: Structure and Principle of Operation

The right-hand prosthesis of a nine-year-old child was created using additive 3D printing technology. The prosthetic parts were printed on a MakerBot Replicator 2 printer (MakerBot Industries, LLC, New York, NY, USA) using MakerBot Desktop 3.10.1 software (MakerBot Industries, LLC, New York, NY, USA). The material used for the structural elements of the prosthesis is PLA (polylactic acid). PLA is one of the most commonly used materials in 3D printing due to its properties, such as adequate strength and rigidity. Since the material is also non-toxic, it is safe for users, especially since the prosthesis elements have direct contact with the body. PLA is also biodegradable, which means its environmental impact is minimal. Although PLA is not as durable as other materials used in 3D printing, such as ABS, Nylon, or ASA, it offers sufficient strength and rigidity for the developed prosthesis. The prosthesis structure of PLA is strong enough to withstand daily use while remaining lightweight and comfortable for the user. The lightweight construction is significant for children with difficulty wearing heavier, conventionally made prostheses. Additionally, additive 3D printing technology allows for the rapid and precise creation of prostheses that can be easily customized to users’ individual needs. This method enables quick modifications to the design and testing of different prototype variants without creating new molds or tools, significantly reducing the time and costs associated with the prototyping process. Thus, 3D printing in prosthesis creation opens up new possibilities in medicine, enabling the creation of personalized and functional solutions for patients of all ages.

The limited mobility of the child’s fingers means that the child cannot perform the precise movements necessary to control traditional mechanical prostheses. In such cases, alternative control methods are necessary, such as electronic and sensory systems that can respond to minimal movements through touch. The absence of metacarpophalangeal joints further complicates the situation, as these joints are crucial for performing complex hand movements. In prostheses for children with such limitations, mechanisms that mimic natural hand movements are often used but controlled by other body parts or external devices. An immobilized wrist joint prevents flexion and extension movements, essential for many daily activities. In such cases, prostheses must be designed to compensate for the lack of wrist mobility, for example, by using joints or rotational mechanisms in other parts of the prosthesis. Thanks to 3D printing, it is possible to quickly change the design and test different variants, significantly reducing the time and costs associated with the prototyping process. Using modern technologies to design prostheses for children with limited mobility opens up new possibilities and gives hope for improving the quality of life for young patients. The movement of individual parts of the hand prosthesis is controlled using touch buttons placed on the child’s vest near the hand (Figure 2).

This innovative solution allows for precise prosthesis control, even with limited natural hand mobility. The touch panel allows for the control of individual prosthesis segments, allowing for various movements. Additionally, the panel enables the programming of the prosthesis movement sequences most frequently used daily (Figure 3). This allows the child to easily and quickly perform complex tasks.

Programmed movement sequences allow for the quick and efficient performance of daily activities, increasing the child’s independence. The ability to program movement sequences allows the prosthesis to be tailored to the individual needs and preferences of the user. Using a touch panel to control the prosthesis is an example of how modern technologies can significantly improve the quality of life for people with disabilities, enabling them to achieve greater independence and comfort in daily functioning.

The development of prosthetic movement control includes the possibility of implementing an EMG sensor subsystem, from which the signal transmitted to the controller will activate the actuators [32]. At a later stage of adapting the prosthesis to the user, this will ensure that the range of movements performed is theoretically limited only by the construction of the prosthesis and not by the capabilities of the control system (Table 1).

Touch buttons can be used to control various functions of the prosthesis, such as bending and straightening fingers, rotating the wrist, or grasping objects (Figure 4). This enables the child to perform many daily activities, such as eating, carrying objects, or playing, with greater ease and independence.

### 2.3. Mechanism and Actuators

The prosthesis is mounted on the child’s back using a vest, which ensures an even distribution of the prosthesis’s weight while providing the freedom of arm movement. This solution allows the child to move their hands naturally, without the restrictions typically associated with wearing a prosthesis. A counterweight placed on the left side of the child’s body is necessary to evenly distribute the weight across the child’s shoulders, which also increases the comfort of wearing the prosthesis. The total weight of the prosthesis, including the controller and battery, which are placed on the back, is 2.5 kg, making it an optimal solution for the young user.

The prosthesis contains eleven electric drives in the form of servomechanisms, including six servomechanisms integrated directly into the prosthetic hand, allowing for the execution of complex movements or sequences of pre-programmed movements. The bending and straightening of the prosthetic fingers are made possible by tendons connecting the servomechanisms and the phalanges. The drives are based on model servomechanisms, ensuring the low weight of the drive components and sufficient torque to manipulate and lift objects with the prosthesis. The servos placed in the hand allow for movement using the following torques described in Table 2.

For the flexion and extension of the arm, worm gear with a 1:72 ratio was added to the standard servomechanism, increasing the torque at the expense of movement speed. Additionally, using such gear enabled the self-locking of the controlled segment, which also translates to a lower current load on the servomechanism when maintaining the set position of the prosthetic arm.

The controller for operating the prosthesis is placed on the back of the vest, along with the necessary components. The system is based on three Atmega 328p microcontrollers (Microchip Technology Inc., 2355 W. Chandler Blvd., Chandler, AZ, USA). The first one controls the angular position of the servos, based on instructions sent by the second microcontroller, which manages the touch panel on the vest near the child’s hand. The third microcontroller performs a control function over the prosthesis’s power supply, with its functions including the following:-Safe start function: The allowed voltage levels on the controllers and battery are checked. Only after confirming proper communication between the controllers and voltage levels does the prosthesis start and position itself in the starting position (with the prosthesis freely aligned along the body).-Battery voltage control: If the threshold voltage value is exceeded, the prosthesis automatically shuts down and signals its status with an audible alert.

The entire prosthesis is powered by a lithium-polymer battery with a nominal voltage of 7.4 V and a capacity of 6000 mAh (Redox One Ltd., Konrad-Adenauer-Allee 11, Dortmund, Germany), providing up to 6 h of operation. The control program applications were written in C++, allowing for precise and reliable prosthesis control. Thanks to a microcontroller, the prosthesis’s operation is stable and reliable, crucial for ensuring user comfort, safety, and functionality (Figure 5).

### 2.4. Experimental Methodology

The study of the pressure force distribution on the ground during walking was conducted using the FreeMed EXTREME Maxi (Sensor Medica SRL, Guidonia Montecelio, Italy) 240 × 50 cm baropodometric platform [33]. The sampling frequency was set at 200 Hz for dynamic testing. The patient underwent proper preparation and familiarization with the platform to ensure the tests were conducted effectively. Additionally, during the dynamic measurements, three measurements were taken for each configuration to obtain precise and reliable data. Three main measurements were taken in the following configurations:(a)Static measurement:
-Free stance without the prosthesis;-Free stance with the prosthesis;-Free stance with the prosthesis without counterbalance.

(b)Dynamic measurement:
-Without the prosthesis (Figure 6A);-With the prosthesis (alongside the torso with a counterbalance);-With the prosthesis (elbow flexed and brought to the torso with a counterbalance) (Figure 6B);-With the prosthesis (alongside the torso without a counterbalance);-With the prosthesis (elbow flexed and brought to the torso without a counterbalance) (Figure 6C).


**Figure 6 jcm-14-02245-f006:**
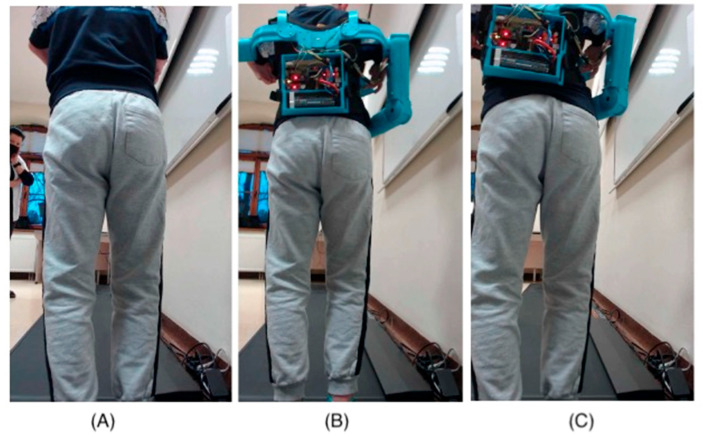
Gait measurement: (**A**) without the prosthesis. (**B**) prosthesis with counterweight (elbow flexed and brought to the torso). (**C**) prosthesis without counterweight (elbow flexed and brought to the torso).

An optical system (2 cameras) was used during the examination.

## 3. Results

Table 3 presents the numerical values of the static test. The load report from the static test without the prosthesis showed that the center of pressure of the left and right feet is not aligned (Figure 7A,B). Only in the case of the prosthesis without a counterbalance are the pressure centers of the individual parts balanced (Figure 7C). The results indicate excessive flatfoot. In the test without the prosthesis, the distribution of load between the forefoot and hindfoot on both feet showed excessive forward loading. The use of the prosthesis resulted in measurements within normal ranges. In all cases, the contact surface areas of the feet with the ground were different, with greater load on the left foot. Between the two contact areas (points of contact) of the forefoot with the ground, there were moderate differences, with the left side being greater. Similarly, between the two contact areas (points of contact) of the hindfoot with the ground, there were excessive differences, again with the left side being larger.

Figure 8A illustrates the dynamic analysis of the gait cycle without prothesis. The time of the initial contact of the foot with the ground averages 23 ms (2% of the gait phase in support). The time for the loading response was 93 ms for the left leg and 88 ms for the right leg (10%). The mid-stance phase lasted 231 ms for the left leg and 221 ms for the right leg (30%). The terminal stance phase was 231 ms for the left leg and 226 ms for the right leg (50%). The results indicate no significant difference between the left and right limbs, as the variations in phase durations are minimal and fall within a comparable range.

Figure 8B illustrates the gait dynamics with the prosthesis. The upper limb remained aligned along the torso. Compared to the trial without the prosthesis, the average duration of the initial contact phase increased to 24 ms. The loading response phase lasted longer for the left leg (102 ms) and shorter for the right leg (82 ms). Similarly, the mid-stance phase was extended, lasting 255 ms for the left leg and 248 ms for the right leg. The terminal stance phase also showed a longer duration, with 258 ms for the left leg and 248 ms for the right leg. A noticeable increase in the foot–ground contact time was evident during the last two phases compared to gait without the prosthesis.

In dynamic testing without the prosthesis, the total surface area of the L—left foot—(71 cm^2^) is smaller than that of the R—right foot (76 cm^2^). The length of the left and right feet’s push-off is the same, measuring 190 mm (Table 4). The average distribution of pressure forces shows significant differences between the left and right feet (average pressure LF = 757 g/cm^2^ RF = 702 g/cm^2^). The maximum load point on the left foot was 1768 g/cm^2^, while on the right foot, it was 1960 g/cm^2^. The distribution of pressure forces between the forefoot and hindfoot on the left foot is within physiological values (forefoot = 61%; hindfoot = 39%), as is that for the right limb (forefoot = 64%; hindfoot = 36%). The medial–lateral distribution of pressure forces on the left foot is 47% on the lateral side and 53% on the medial side, while on the right foot, it is 74% on the lateral side and 26% on the medial side. The contact time with the ground for the left foot was measured at 690 ms and for the right foot at 673 ms (Figure 9).

In dynamic testing with the prosthesis and the limb aligned with the torso, the total surface area of the left foot (64 cm^2^) is smaller than that of the right foot (68 cm^2^). The length of the left foot’s push-off is 190 mm, while the right foot’s is 170 mm. The average distribution of pressure forces shows significant differences between the left and right feet (average pressure LF = 852 g/cm^2^ RF = 619 g/cm^2^). The maximum load point on the left foot was 1932 g/cm^2^, while on the right foot, it was 1564 g/cm^2^. The distribution of pressure forces between the forefoot and hindfoot on the left foot is within physiological values (forefoot = 61%; hindfoot = 39%), whereas the anterior–posterior distribution of pressure forces on the right foot does not fall within the norm (forefoot = 71%; hindfoot = 29%).

The medial–lateral distribution of pressure forces on the left foot is 75% on the lateral side and 25% on the medial side, while on the right foot, it is 73% on the lateral side and 27% on the medial side (Figure 10). The contact time with the ground for the left foot was measured at 766 ms and for the right foot at 735 ms.

In the test with the prosthesis and counterweight (elbow bent), the total surface area of the left foot (62 cm^2^) is smaller than that of the right foot (78 cm^2^). The length of the left foot’s push-off is 160 mm, while the right foot’s is 180 mm. The average distribution of pressure forces shows no significant differences between the left and right feet (average pressure LF = 738 g/cm^2^ RF = 748 g/cm^2^). The maximum load point on the left foot is 1688 g/cm^2^, while on the right foot, it is 1784 g/cm^2^. The distribution of pressure forces between the forefoot and hindfoot on the left foot is within physiological values (forefoot = 57%; hindfoot = 43%), whereas the anterior–posterior distribution of pressure forces on the right foot does not fall within the norm (forefoot = 65%; hindfoot = 35%). The medial–lateral distribution of pressure forces on the left foot is 42% on the lateral side and 58% on the medial side, while on the right foot, it is 36% on the lateral side and 64% on the medial side. The contact time with the ground for the left foot was measured at 759 ms and for the right foot at 721 ms (Figure 11).

In the test without the counterweight (limb aligned with the torso), the total surface areas of the feet are similar (LF = 64 cm^2^ RF = 64 cm^2^). The length of the left foot’s push-off is 190 mm, while the right foot’s is 170 mm. The average distribution of pressure forces shows significant differences between the left and right feet (average pressure LF = 582 g/cm^2^ RF = 898 g/cm^2^). The maximum load point on the left foot is 1520 g/cm^2^, while on the right foot, it is 1952 g/cm^2^. The distribution of pressure forces between the forefoot and hindfoot on the left foot shows that the ratio falls outside the normal range (forefoot = 54%; hindfoot = 46%), whereas the anterior–posterior distribution of pressure forces on the right foot is within the norm (forefoot = 62%; hindfoot = 38%). The medial–lateral distribution of pressure forces on the left foot is 68% on the lateral side and 32% on the medial side, while on the right foot, it is 51% on the lateral side and 49% on the medial side. The contact time with the ground for the left foot was measured at 813 ms and for the right foot at 713 ms (Figure 12).

The recent test without the counterweight, with the prosthesis bent at the elbow joint, showed that the total surface areas of the feet are similar (LF = 71 cm^2^ RF = 72 cm^2^). The length of the left foot’s push-off is 180 mm, while the right foot’s is 190 mm. The average distribution of pressure forces shows significant differences between the left and right feet (average pressure LF = 595 g/cm^2^ RF = 681 g/cm^2^). The maximum load point on the left foot is 1796 g/cm^2^, while on the right foot, it is 1588 g/cm^2^. The distribution of pressure forces between the forefoot and hindfoot on the left foot shows that the ratio falls outside the normal range (forefoot = 65%; hindfoot = 35%), while the anterior–posterior distribution of pressure forces on the right foot is within the norm (forefoot = 57%; hindfoot = 43%). The medial–lateral distribution of pressure forces on the left foot is 58% on the lateral side and 42% on the medial side, while on the right foot, it is 66% on the lateral side and 34% on the medial side. The contact time with the ground for the left foot was measured at 865 ms and for the right foot at 771 ms (Figure 13).

## 4. Discussion

The gait analysis conducted on an individual with TAR syndrome, with and without an upper limb prosthesis, provided valuable insights into the biomechanical differences between the left and right lower limbs under various conditions. The data collected in the dynamic testing process helped evaluate foot pressure distribution, surface area, push-off length, and contact time with the ground, contributing to a better understanding of the gait alterations resulting from prosthetic use.

In all test conditions, the total surface area of the left foot was consistently smaller than that of the right foot. This difference could be attributed to the individual’s anatomical variations related to TAR syndrome, which typically involves the underdevelopment or absence of the radius, affecting limb alignment and overall body symmetry. Despite this, the length of the push-off for both feet in each condition was nearly identical (190 mm for the left and 170 mm for the right), indicating that, at least in terms of the push-off phase, the individual maintained relatively balanced mechanics. The consistency in push-off length suggests an adaptive strategy to compensate for differences in limb structure, particularly when using the prosthesis.

The distribution of pressure forces showed notable differences between the left and right foot, especially in the condition with the prosthesis and counterweight. In these tests, the average pressure on the left foot (738 g/cm^2^ to 852 g/cm^2^) was generally higher than that on the right foot (619 g/cm^2^ to 748 g/cm^2^), suggesting that the individual may rely more on the left foot during the stance and push-off phases when the prosthetic limb is in use. These differences could be due to altered body weight distribution or compensatory mechanisms driven by the lack of full functionality in the prosthetic arm, which might lead to increased load-bearing on the left leg. In the condition with the prosthesis and counterweight, the maximum load points on the left foot (1688–1932 g/cm^2^) were also higher than those on the right foot (1564–1784 g/cm^2^), reflecting an increased reliance on the left lower limb, possibly to maintain stability and balance. The asymmetry in pressure distribution could also be due to the altered gait mechanics induced by the use of the upper limb prosthesis, which may affect the overall alignment and biomechanics of the lower limbs.

The forefoot–hindfoot pressure distribution showed that both the left and right feet maintained pressure within physiological values in most test conditions. In the left foot, the forefoot pressure was higher (61–65%) compared to the hindfoot, suggesting a more pronounced use of the forefoot during the stance phase, which might be due to compensation for reduced force generation in the upper limbs. This may lead to a more forward-oriented posture and gait pattern to minimize reliance on the prosthetic limb. On the other hand, the right foot displayed a more varied pattern, with some tests showing pressure ratios outside the physiological norm, particularly in the conditions with the prosthesis and counterweight. This indicates potential compensatory mechanisms in the right limb that could be a response to the asymmetric load-bearing caused by the use of the prosthesis.

Medial–lateral pressure distribution further illustrates how the individual adapts to the use of the prosthesis. The left foot showed a more balanced distribution of pressure between the medial and lateral sides in most conditions, with the lateral side bearing more weight (47–75%). In contrast, the right foot exhibited a higher concentration of pressure on the lateral side (66–74%), which suggests an altered gait pattern with a tendency to shift more weight toward the outer edge of the right foot. This may be indicative of an attempt to reduce load on the prosthetic side or an adaptation to prevent the overloading of the left limb. Such a lateral shift could increase the risk of strain on the foot and ankle joints, potentially leading to longer-term musculoskeletal issues.

The contact time with the ground for the left foot was consistently longer (690–865 ms) than that for the right foot (673–771 ms), with the largest difference observed when the counterweight was used. This extended contact time for the left foot may indicate that the individual relies more on the left side to maintain balance during stance, possibly as a compensation for the asymmetry created by the prosthetic limb. The reduced contact time for the right foot, especially when using the prosthesis, may reflect a quicker push-off or reduced load-bearing during gait, which could be linked to the altered mechanics resulting from prosthesis usage.

There are some limitations of this study. The small sample size focuses on a single 10-year-old patient, making it difficult to generalize the findings to a larger population of individuals with congenital upper limb deformities. There is no comparison group of individuals without limb deformities, which limits the ability to determine the specific effects of the deformity on posture and gait. Short-term observation provides only a snapshot of the patient’s posture and gait at a specific time. Long-term effects, such as adaptations over time, are not assessed. Limited motion analysis based on the baropodometric platform only measures pressure distribution and does not capture joint kinematics or muscle activation patterns, which are crucial for understanding compensatory movements. There may be natural variability in walking patterns due to fatigue, psychological factors, or inconsistencies in prosthetic use, which can influence the results. Despite these limitations, this study provides valuable insights into posture and gait mechanics in a child with a congenital upper limb deformity.

## 5. Conclusions

The results of the gait analysis demonstrate how an individual with TAR syndrome adapts to the challenges of using an upper limb prosthesis, with the left foot bearing more pressure and experiencing longer contact times. The observed asymmetries in pressure distribution and foot surface area highlight the impact of both anatomical differences and prosthetic use on gait mechanics. Rehabilitation strategies, such as the SPIDER tool, which has shown promise in enhancing motor control and stability in patients with neurological disabilities, could be explored for their potential to improve gait symmetry in individuals with TAR syndrome [28].

Future research should include a larger sample size, a control group, kinematic and muscle activity analysis, and long-term adaptation studies to enhance the reliability and applicability of the findings. Further studies with larger sample sizes and advanced imaging techniques are necessary to better understand the long-term effects of these adaptations on joint health and overall mobility. These insights can guide the development of more effective rehabilitation strategies for individuals with similar conditions, focusing on balancing load distribution, improving gait dynamics, and minimizing the risk of compensatory injuries.

## Figures and Tables

**Figure 2 jcm-14-02245-f002:**
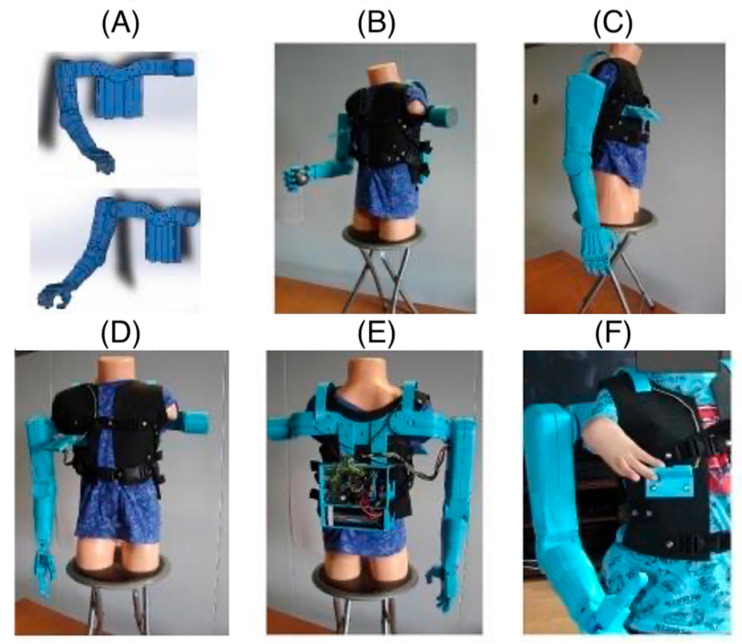
Photos of the prosthesis hand: (**A**) 3D model, (**B**–**E**) prosthesis hand, (**F**) denture movement control by the user.

**Figure 3 jcm-14-02245-f003:**
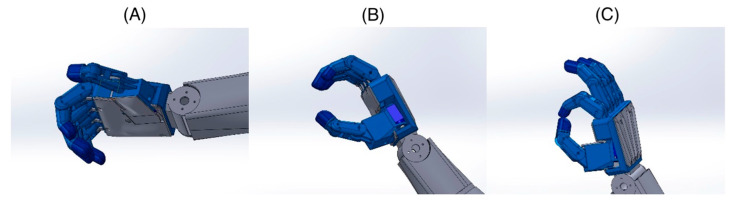
Example types of hand grips for the prosthesis: (**A**) door handle grip, (**B**) basic grip for various objects, (**C**) precision pinch grip.

**Figure 4 jcm-14-02245-f004:**
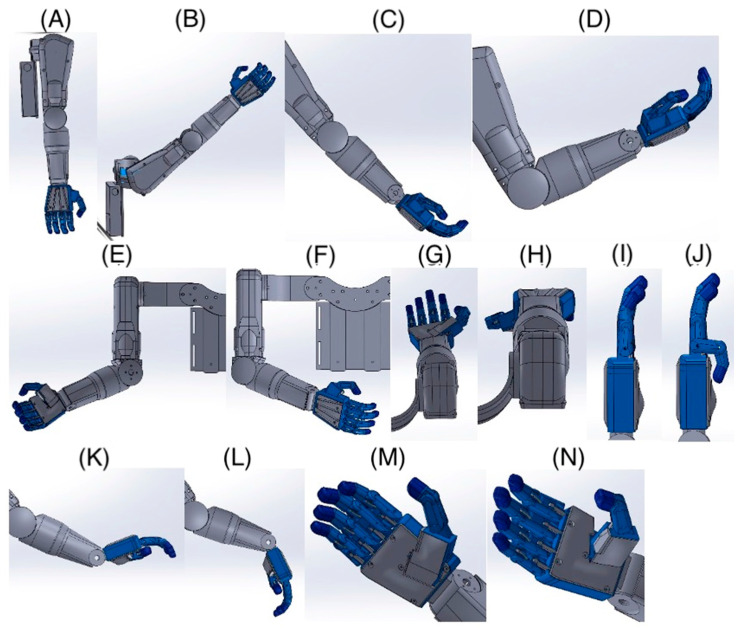
Possible movements of the prosthesis: (**A**) arm extension, (**B**) arm flexion, (**C**) elbow extension, (**D**) elbow flexion, (**E**) arm external rotation, (**F**) arm internal rotation, (**G**) forearm supination, (**H**) forearm pronation, (**I**) little finger extension, (**J**) little finger flexion, (**K**) wrist extension, (**L**) wrist flexion, (**M**) thumb adduction, (**N**) thumb abduction.

**Figure 5 jcm-14-02245-f005:**
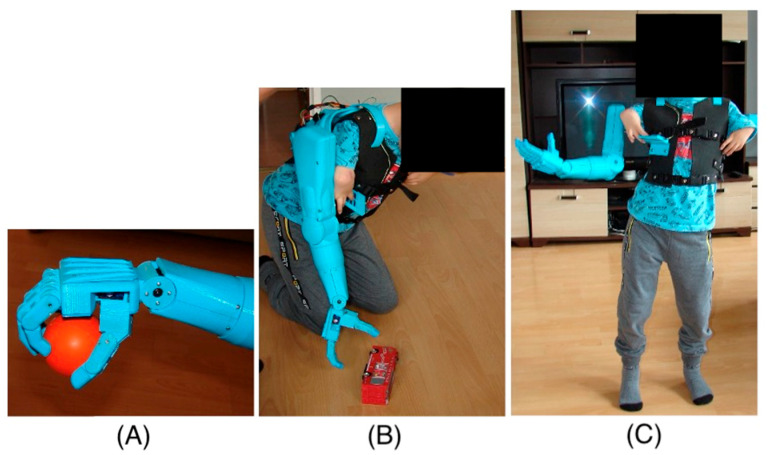
Prosthesis function tests: (**A**) ball grip, (**B**) picking up a toy from the ground, (**C**) the tested kinematic movements of the prosthesis.

**Figure 7 jcm-14-02245-f007:**
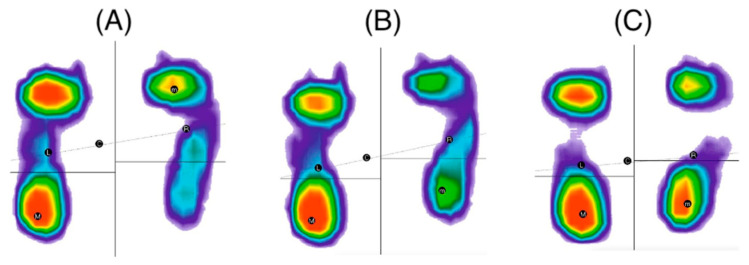
Images of static examination of feet in free position: (**A**) without prosthesis; (**B**) with prosthesis—limb directed along trunk; (**C**) with prosthesis without counterweight.

**Figure 8 jcm-14-02245-f008:**
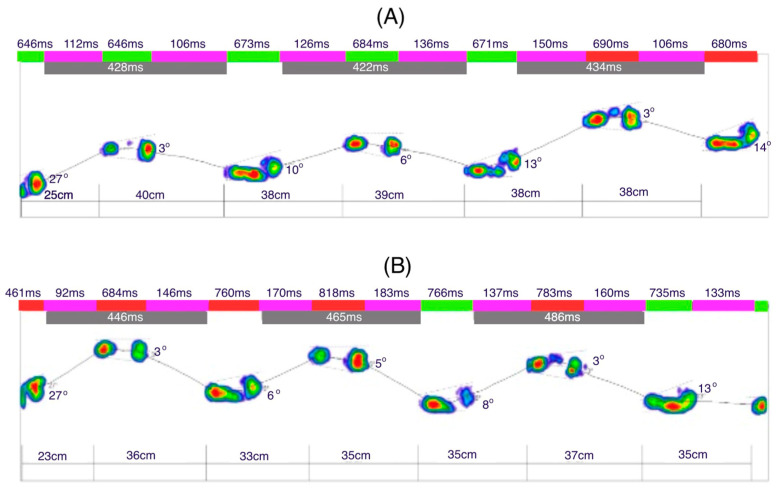
Gait dynamics without prothesis (**A**) Gait dynamics with prothesis (**B**) (red: left leg; green: time right leg; purple: double support; gray: swing time [ms]).

**Figure 9 jcm-14-02245-f009:**
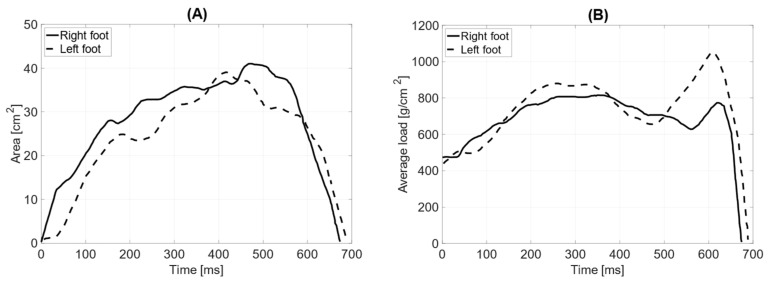
Gait parameters without the prosthesis: (**A**) surface area during the foot contact phase with the ground [cm^2^]; (**B**) average load [g/cm^2^].

**Figure 10 jcm-14-02245-f010:**
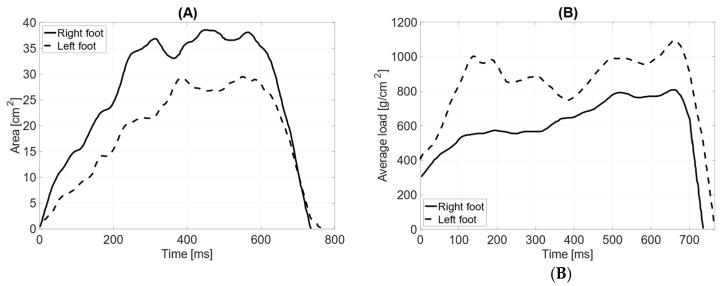
Gait parameters with the prosthesis and counterweight (upper limb along the trunk): (**A**) surface area during the ground contact phase [cm^2^]; (**B**) average load [g/cm^2^].

**Figure 11 jcm-14-02245-f011:**
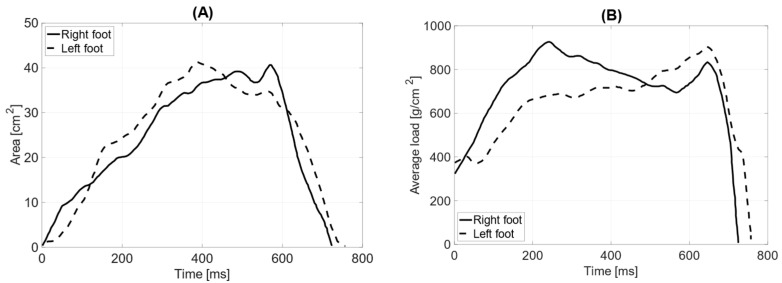
Gait parameters with the prosthesis and counterbalance (limb at the trunk, flexed at the elbow): (**A**) surface area during the ground contact phase [cm^2^]; (**B**) average load [g/cm^2^].

**Figure 12 jcm-14-02245-f012:**
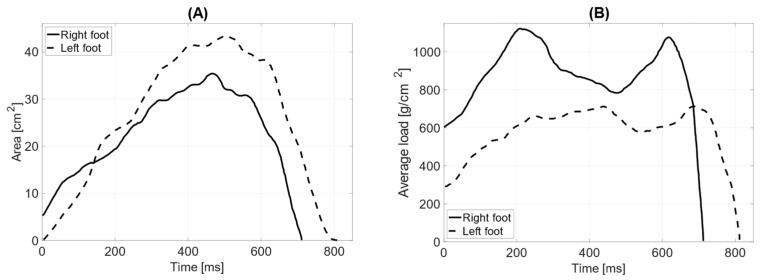
Gait parameters with the counterbalanced prosthesis (limb along the trunk): (**A**) surface area during the ground contact phase [cm^2^]; (**B**) average load [g/cm^2^].

**Figure 13 jcm-14-02245-f013:**
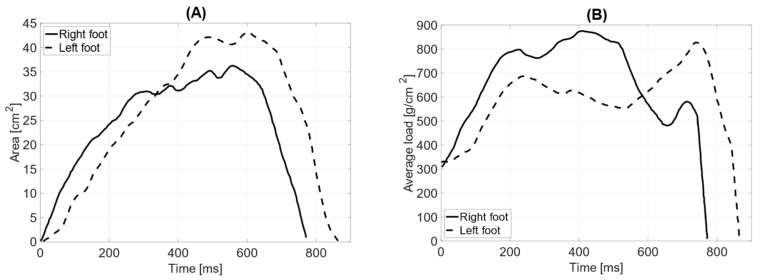
Gait parameters with the prosthesis without a counterweight (flexed limb): (**A**) surface area during the ground contact phase [cm^2^]; (**B**) average load [g/cm^2^].

**Table 1 jcm-14-02245-t001:** The kinematic parameters of the prosthesis.

Part	Type of Movement	Range of Motion [Deg]
Fingers	Flexion/Extension	90
Thumb	Adduction/Abduction	90
Wrist	Flexion/Extension	65
Forearm	Pronation/Supination	−90/+90
Elbow	Flexion/Extension	80
Shoulder	Medial/Lateral rotation	−60/+45
Shoulder	Flexion/Extension	−45/+60

**Table 2 jcm-14-02245-t002:** The dynamic parameters of the prosthesis.

Part of Hand Prosthesis	Movement of Prosthesis	Torque [Nm]
Five fingers	Independent bending/straightening	0.34
Thumb	Adduction/Abduction	0.59
Wrist	Flexion/Extension	2.45
Forearm	Rotation	2.45
Elbow	Flexion/Extension	3.43
Shoulder	Rotation	3.43
Shoulder	Flexion/Extension	82

**Table 3 jcm-14-02245-t003:** The numerical values of the static test.

	Parameter	Without a Prosthesis		Prosthesis with Counterweight		Prosthesis Without Counterweight	
		Left	Right	Left	Right	Left	Right
Forefoot	Area [cm^2^]	45	40	48	41	37	30
	Load [%]	34	26	29	21	25	16
	Weight factor R/F [%]	54	70	46	56	43	40
Hindfoot	Area [cm^2^]	28	22	31	27	30	30
	Load [%]	29	11	34	16	34	25
	Weight factor R/F [%]	46	30	54	44	57	60
All	Area [cm^2^]	73	62	78	68	67	60
	Load [%]	63	37	63	37	59	41
	Max. load [g/cm^2^]	673	464	661	328	724	539
	Mean load [g/cm^2^]	259	179	242	163	264	205

**Table 4 jcm-14-02245-t004:** The numerical values of the dynamic test (L—left; R—right).

	Without Prothesis		Prosthesis—Limb Directed Along the Trunk		Counterweight Prosthesis Attached to the Waist		Prosthesis Without Counterweight Along the Trunk		Prosthesis Without Counterweight Attached to the Waist	
	L	R	L	R	L	R	L	R	L	R
Max load [g/cm^2^]	1768	1960	1932	1564	1784	1688	1520	1952	1796	1588
Mean load [g/cm^2^]	757	702	852	619	660	738	582	898	595	681
Area [cm^2^]	71	76	64	68	70	62	64	64	71	72
Plantar axis [deg]	3	−10	0	−13	−3	−2	−7	−13	−11	−10
Foot length [mm]	190	190	190	170	200	160	190	170	180	190
Delta CoF [mm]	160	170	180	148	154	133	167	135	135	173
Forefoot load R/F [%]	61	64	61	71	63	58	54	62	65	57
Hindfoot load R/F [%]	39	36	39	29	37	42	46	38	35	43
Middle load L/L [%]	53	26	25	27	24	43	32	49	42	34
Side load L/L [%]	47	74	75	73	76	57	68	51	58	66

## Data Availability

Data available on request: sebastian.glowinski@upsl.pl.

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
