# Peer review of "Gait Analysis with an Upper Limb Prosthesis in a Child with Thrombocytopenia–Absent Radius Syndrome"

_jcm, 2025, doi:10.3390/jcm14072245_

Round 1
Reviewer 1 Report
Comments and Suggestions for Authors
- in the abstract, the methods are actually the aim. Please revise.
- the introduction section is well writen, with recent and relevant references
- can the authors provide the informed consent form, and ethical approval number, and date, together with the institution, for the case?
- restructure information from row 205 - row 213 in a Table in a better visible format.
- the discussions are well written and concise
- conclusions are clear.
- References are up to date and relevant.
- the title should make clear that this is a case report. This is a lower level of evidence, and that should be known from the title.
Author Response
Dear Reviewer,
The authors would like to thank the reviewer editor for his effort in reviewing the manuscript and for his valuable and constructive comments and fruitful observations, which helped in improving the quality of the manuscript to a publishable standard. Below are the responses to the reviewer comments and suggestions.
Point 01. In the abstract, the methods are actually the aim. Please revise.
Response 01. Thank you for your valuable note. The abstract was revised, and the following sentence was edited. The methodology is based on a study that evaluated the gait parameters of a 10-year-old child with TAR syndrome under static and dynamic conditions, both with and without the use of a custom-designed upper limb prosthesis. The analysis focused on assessing the prosthesis's impact on gait symmetry and biomechanics. A key aspect of the methodology involved studying the distribution of pressure forces on the ground during walking using the FreeMed EXTREME Maxi baropodometric platform
Point 02. The introduction section is well writen, with recent and relevant references
Response 02. Thank you for your positive feedback. We are glad to hear that you found the introduction well written and the references relevant and up to date.
Point 03. Can the authors provide the informed consent form, and ethical approval number, and date, together with the institution, for the case?
Response 03. Yes of course. The study was conducted in accordance with the Declaration of Helsinki, and the study protocol was approved by the Bioethics Committee at the district medical chamber in Gdansk (KB-14/20). The informed consent form was sent to Associate Editor Journal of Clinical Medicine.
Point 04. Restructure information from row 205 - row 213 in a Table in a better visible format.
Response 04. Thank you for this remark. According to your suggestion, the following table was added to the text.
Table 2. Dynamic parameters of the prosthesis
|
Part of hand prosthesis |
Movement of prosthesis |
Torque [Nm] |
|
Five fingers Thumb Wrist Forearm Elbow Shoulder Shoulder |
Independent bending/straightening Adduction / Abduction Flexion / Extension Rotation Flexion / Extension Rotation Flexion / Extension |
0.34 0.59 2.45 2.45 3.43 3.43 82 |
Point 05. The discussions are well written and concise. Conclusions are clear. References are up to date and relevant.
Response 05. Thank you for your positive feedback. We appreciate your comments on the clarity and conciseness of our discussions, as well as the relevance of our conclusions and references.
Point 05. The title should make clear that this is a case report. This is a lower level of evidence, and that should be known from the title
Response 06. According to the reviewer suggestion the title was revised in the following form:
“Gait Analysis with Upper Limb Prosthesis in a Child with Thrombocytopenia Absent Radius Syndrome”.
Reviewer 2 Report
Comments and Suggestions for Authors
Thank you very much for the opportunity to review extremely interesting studies. Please provide additional information:
1. Please provide additional information about the locomotion tests: was there and what was the patient's preparation/familiarization with the platform? Did the patient have a triple measurement in each configuration during dynamic measurement?
2. Why was no statistical analysis comparing the right and left side (Table 2) performed?
3. The Discussion lacks analysis and interpretation of the results in comparison with studies available in the literature.
4. Please provide Study Limitation.
Author Response
Dear Reviewer,
The authors would like to thank the reviewer for his effort in reviewing the manuscript and for his valuable and constructive comments and fruitful observations, which helped in improving the quality of the manuscript to a publishable standard. Below are the responses to the reviewer comments and suggestions.
Point 01. Thank you very much for the opportunity to review extremely interesting studies.
Response 01. Thank you for your kind words and for taking the time to review our work. We truly appreciate your valuable feedback.
Point 02. Please provide additional information about the locomotion tests: was there and what was the patient's preparation/familiarization with the platform? Did the patient have a triple measurement in each configuration during dynamic measurement?
Response 02. Thank you to the reviewer for this insightful and valuable comment. Yes, the patient underwent proper preparation and familiarization with the platform to ensure the tests were conducted effectively. Additionally, during the dynamic measurements, three measurements were taken for each configuration to obtain precise and reliable data. The remark was added to the text in subsection 2.5. Experimental Methodology.
Point 03. Why was no statistical analysis comparing the right and left side (Table 2) performed?
Response 03. Thank you for this precise remark. The analysis focused on a single patient with the rare and specific condition of TAR syndrome, which significantly limited the possibility of conducting a statistical analysis. The authors determined that a larger study sample would be necessary for reliable statistical comparisons. In this context, direct comparisons of values presented in Table 3 allowed for evaluating the static trial results in the upright posture. This approach enabled meaningful insights within the methodological constraints.
Point 04. The Discussion lacks analysis and interpretation of the results in comparison with studies available in the literature.
Response 04. Thank you for being so considerate. We would like to emphasize that the available scientific literature on TAR syndrome focuses primarily on other medical and biomechanical aspects, such as diagnostics, surgical treatment or functionality of the upper limb. However, there are no documented cases of gait analysis in children with TAR. Our study explicitly examines the unique impact of a dedicated upper limb prosthesis on gait biomechanics, filling a gap in existing research. This analysis can serve as a basis for future research that includes comparisons with the literature in areas more closely related to our work. We would appreciate this comment, as it allows us to expand on this topic in future publications.
Point 05. Please provide Study Limitation.
Response 05.
Thank You for final valuable note. According to the suggestion the following summarising remarks are added in the Conclusions.
There are some limitations of the Study. Small sample size focuses on a single 10-year-old patient, making it difficult to generalize the findings to a larger population of individuals with congenital upper limb deformities. There is no comparison group of individuals without limb deformities, which limits the ability to determine the specific effects of the deformity on posture and gait. Short-term observation provides only a snapshot of the patient’s posture and gait at a specific time. Long-term effects, such as adaptations over time, are not assessed. Limited motion analysis based on the baropodometric platform only, measures pressure distribution and does not capture joint kinematics or muscle activation patterns, which are crucial for understanding compensatory movements. Variability in walking patterns may exhibit natural variability in walking due to fatigue, psychological factors, or inconsistencies in prosthetic use, which can influence the results.
Despite these limitations, the study provides valuable insights into posture and gait mechanics in a child with a congenital upper limb deformity. However, future research should include a larger sample size, a control group, kinematic and muscle activity analysis, and long-term adaptation studies to enhance the reliability and applicability of the findings.
Reviewer 3 Report
Comments and Suggestions for Authors
A good review of absent radius syndrome, and since this is a rare disease this case report seems to be reasonable and it’s writing.
I would only add a limitation in the study to be that a gait analysis map was not used in the future may benefit from a gait analysis map however the authors were very good in putting the limitations of their assistive device and compensate mechanism analyze during gait.
Also, they might consider including other assistive devices available, and why those were not tried in this specific case report. I’ll do it might also be that this child because of the age and their body weight. This was the best advice to use.
Author Response
Dear Reviewer,
The authors would like to thank the reviewer for his effort in reviewing the manuscript and for his valuable and constructive comments and fruitful observations, which helped in improving the quality of the manuscript to a publishable standard. Below are the responses to the reviewer comments and suggestions.
Point 01. A good review of absent radius syndrome, and since this is a rare disease this case report seems to be reasonable and it’s writing;
Response 01. Thank you for your thoughtful review. We appreciate your recognition of our work and the relevance of this case report in highlighting such a rare condition.
Point 02. I would only add a limitation in the study to be that a gait analysis map was not used in the future may benefit from a gait analysis map however the authors were very good in putting the limitations of their assistive device and compensate mechanism analyze during gait.
Response 02.
Thanks for your valuable input and we want to confirm that we are aware of the obvious limitations of the presented studies. Therefore, we have included the following assessment in the Conclusions:
There are some limitations of this study. Small sample size focuses on a single 10-year-old patient, making it difficult to generalize the findings to a larger population of individuals with congenital upper limb deformities. There is no comparison group of individuals without limb deformities, which limits the ability to determine the specific effects of the deformity on posture and gait. Short-term observation provides only a snapshot of the patient’s posture and gait at a specific time. Long-term effects, such as adaptations over time, are not assessed. Limited motion analysis based on the baropodometric platform only, measures pressure distribution and does not capture joint kinematics or muscle activation patterns, which are crucial for understanding compensatory movements. Variability in walking patterns may exhibit natural variability in walking due to fatigue, psychological factors, or inconsistencies in prosthetic use, which can influence the results.
Despite these limitations, the study provides valuable insights into posture and gait mechanics in a child with a congenital upper limb deformity. However, future research should include a larger sample size, a control group, kinematic and muscle activity analysis, and long-term adaptation studies to enhance the reliability and applicability of the findings.
Point 03. Also, they might consider including other assistive devices available, and why those were not tried in this specific case report. I’ll do it might also be that this child because of the age and their body weight. This was the best advice to use.
Response 03.
Thank you for your thoughtful and detailed observation. In this case report, the decision to focus on a single assistive device—an upper limb prosthesis—was informed by the child’s unique physical attributes, including their age, body weight, and overall health condition. Among available options, this prosthesis was deemed the most appropriate and practical solution to address their specific needs and challenges at this stage of development.
We agree that incorporating a comparison with other assistive devices could provide additional insights. However, this case report aimed primarily to explore the child's adaptation to a single device, making it a foundational study for future comparative research. Including alternative devices in future studies could indeed broaden the understanding of their impacts on gait mechanics.
Once again, we appreciate the constructive feedback, as it opens avenues for enriching future investigations and furthering rehabilitation strategies for individuals with conditions like TAR syndrome.
Round 2
Reviewer 2 Report
Comments and Suggestions for Authors
Thank you for taking my comments into account and responding to the opinions. Please place the Study limitation at the end of the Discussion and not in the Conclusions.
Author Response
Dear reviewer,
thank you for your suggestion. The Study of limitation was placed at the end of section Discussion.